# The Experience and Development of the Treatment Technology of Municipal Solid Waste Leachate in China

Xinxin Song [1,2], Haihua Min [3], Lejun Zhao [2], Qingming Fu [4], Wei Zheng [3], Xingjian Wang [3], Ximing Ding [3], Lingjie Liu [1] and Min Ji [1,*]

1   School of Environmental Science and Engineering, Tianjin University, Tianjin 300072, China
2   Tianjin Municipal Engineering Design and Research Institute Co., Ltd., Tianjin 300392, China
3   Zhongcheng Institute (Beijing) Environmental Technology Co., Ltd., Beijing 100120, China
4   Hangzhou Environment Group Co., Ltd., Hangzhou 310022, China
*   Correspondence: jimin@tju.edu.cn; Tel.: +86-22-87402072

**Abstract:** This paper reviews the characteristics of leachate produced from municipal solid waste landfills, incineration plants, transfer stations, and food waste anaerobic fermentation plants. In addition, the development of the leachate treatment technology used in China is investigated. The development period of leachate treatment technology in China can be divided into five stages: the early 1990s with simple biological treatment, the mid-to-late 1990s with ammonia stripping + anaerobic + aerobic treatment, from 2000 to 2008 with a two-stage disc tube reverse osmosis (DT-RO) process, from 2008 to 2015 with an anaerobic + aerobic + advanced treatment process, and from 2015 until the present with a diversified and full quantitative process. Furthermore, under the requirements of ecological environmental protection and "dual carbon" development concepts, this paper advises the future development trend of Chinese municipal solid waste leachate, which will enact more scientific emission standards and local standards, become inevitable for the green ecology of the technical route, be more professional and intelligent for construction and operation, and diversify resource utilization of the treatment facilities process.

**Keywords:** municipal solid waste leachate; experience and development; waste water characteristics; treatment technology; future development trend





## 1. Introduction

The process of development of the leachate treatment industry has corresponded with the development of the Chinese waste treatment industry during different periods. Originally, municipal solid waste disposal in China dates back to the 1980s, and the disposal methods gradually developed from the simple stacking of waste to the modern sanitary landfills [1]. Leachate is a high-concentration organic wastewater produced in the process of garbage collection, transportation, and disposal [2]. Generally, leachate has been mainly divided into landfill leachate, incineration plant leachate, transfer station leachate, and also organic waste anaerobic digested slurry.

With the continuous improvement of the landfill leachate collection system and environmental protection regulations in China, landfill leachate had become a key pollutant in sanitary landfills [3]. These regulations have driven the development of the Chinese landfill leachate treatment industry. With the rapid development of the economy in China, various waste treatment and disposal technologies have also developed rapidly. For example, waste incineration power plants gradually replaced sanitary landfills as a new mainstream waste treatment method [4]. In addition, due to the rapid development of urbanization and the increase of garbage collection and transportation distance, the application of transfer stations had gradually been promoted. Since the high-concentration wastewater collected by waste incineration power plants and transfer stations have similar properties to fresh landfill leachate, these three kinds of wastewater are collectively referred to as leachate. In

2010, four ministries and commissions, including the National Development and Reform Commission and the Ministry of Agriculture of China, issued a notice to select qualified cities or municipalities to carry out pilot work on resource utilization and harmless disposal of kitchen waste. In 2016, China began to generally implement waste classification nation-wide [5]. By the end of 2021, almost all cities in China with a population of more than one million had built kitchen waste recycling facilities [6]. Household food waste is generated during the waste classification process, and its physical composition is similar to that of restaurant kitchen waste. These two types of waste are collectively referred to as food waste. The mainstream process technology route adopted for food waste is the pretreatment + wet or dry anaerobic fermentation process [7]. The digested slurry has similar properties to landfill leachate, and it has also been classified as municipal solid waste leachate [8].

Due to the complexity and uncertainty of landfill leachate, the disposal technology of landfill leachate is very complicated. In order to improve the ecological environment and provide cost-effective treatment options for managers, the objectives of this review are (1) to summarize the historical process of municipal solid waste leachate treatment in China and analyze the characteristics of landfill in different countries, (2) to summarize processes of leachate treatment and discharge standards at different stages in China, (3) and to prospect and advise the technology for leachate treatment and standards establishment for the future.

## 2. Leachate Characteristics and Their Influencing Factors

### 2.1. Landfill Leachate Characteristics and Their Influencing Factors

Landfill leachate is produced in the process of landfill treatment, and its characteristics are affected by various factors, such as the composition of landfill waste, the landfill process, and external environmental conditions. According to the different ages of landfills, landfill leachate can be divided into early landfill leachate (landfill age is within 5 years), mid-term landfill leachate (landfill age is 5 to 10 years old), and late landfill leachate (landfill age is more than 10 years) [9].

This paper takes landfill leachate produced from Hangzhou in southern China and Tianjin in northern China as examples, and the characteristics and analysis results are shown in Table 1. To reduce the impact of newly filled garbage on the sampling results, the sampling points were set at fixed positions in two landfills.

**Table 1.** Leachate characteristics of the Hangzhou and Tianjin landfills (annual average).

| City Name | Sampling Time | pH | $COD_{Cr}$ [1] (mg/L) | $BOD_5$ [2] (mg/L) | $BOD_5/COD_{Cr}$ | $NH_3$-N (mg/L) | Salt Content (mg/L) |
|---|---|---|---|---|---|---|---|
| Hangzhou | 2007 (early stage) | 6.29 | 13,200 | 5940 | 0.45 | 580 | 8960 |
| | 2015 (mid-term) | 7.13 | 7650 | 1950 | 0.25 | 1890 | 9780 |
| | 2021 (late stage) | 7.82 | 3840 | 350 | 0.09 | 2380 | 10,120 |
| Tianjin | 2010 (early stage) | 6.08 | 23,680 | 11,300 | 0.48 | 1175 | 13,150 |
| | 2015 (mid-term interim) | 6.76 | 11,100 | 3230 | 0.29 | 1425 | 14,900 |
| | 2021 (late stage) | 7.49 | 6600 | 1280 | 0.19 | 1840 | 13,700 |

Note(s): [1] $COD_{Cr}$, chemical oxygen demand based on the potassium dichromate method. [2] $BOD_5$, biochemical oxygen demand for 5 days.

As shown in Table 1, with the increase in landfill age, the concentrations of $COD_{Cr}$ and $BOD_5$ of the leachate decreased gradually. The $BOD_5/COD_{Cr}$ value decreased from 0.48 to 0.09, indicating that the biodegradability was reduced. The C/N ratio of the late landfill leachate was unbalanced. The values of pH, ammonia nitrogen, electrical conductivity,

etc., in the landfill had an upward trend. The differences in landfill leachate characteristics between northern and southern China were influenced by types of landfill waste, amount of precipitation and snowfall, dietary habits, and refinement management mode [10]. Generally, leachate organic load concentration was lower in southern cities due to heavier rainfall. Furthermore, these characteristics were more obvious in overaged landfills. At the same time, due to the relatively high temperature in South China, the leachate characteristic period was significantly shortened.

On 31 July 2020, the Chinese National Development and Reform Commission, the Ministry of Housing and Urban–Rural Development, and the Ministry of Ecology and Environment jointly issued the "Implementation Plan for Strengthening Weaknesses in Urban Domestic Waste Classification and Treatment Facilities". This scheme requires that primary municipal solid waste must essentially achieve a "zero landfill" by 2023. To date, the cities with a population of more than one million in China have basically achieved a "zero landfill" by incineration treatment, and landfills previously in service are also in a state of simple closure. Thus, the Ministry of Housing and Urban–Rural Development promulgated a technical standard of the People's Republic of China for the ecological restoration of old municipal solid waste landfill sites. This indicates that the municipal solid waste landfill will formally withdraw from the historical stage.

Due to ecological restoration, the role of the entire external conditions of the landfill has fundamentally changed, and there are essential differences between Chinese landfills and those in Europe and the United States. The more than 10-year landfill leachate characteristics of various countries are shown in Table 2.

**Table 2.** Leachate characteristics in late-stage landfills (more than 10 years) of different countries.

| Country | pH | $COD_{Cr}$ (mg/L) | $BOD_5$ (mg/L) | $BOD_5/COD_{Cr}$ | $NH_3$-N (mg/L) | Salt Content (mg/L) | Age | References |
|---|---|---|---|---|---|---|---|---|
| China | 7.5–9.0 | 500–6000 | 100–550 | 0.01–0.1 | 100–4500 | 2000–12,000 | 15 | |
| USA | - | 100–350 | - | - | 30–55 (TN [1]) | - | >13 | [11] |
| Japan | - | 300 | 20 | 0.07 | 500 | - | 16 | [12] |
| Belgium | 8.2–8.8 | 645–1230 | 78–213 | 0.12–0.17 | 255–648 | 9.1–11.2 (EC [2]) | 30 | [13] |
| Columbia | 7.94–9.3 | 23,000–35,000 | 2700–4000 | 0.11–0.12 | - | - | >10 | [14] |
| India | 7.3–9.3 | 72–5100 | 3–207 | 0.04 | - | - | 16 | [15] |

Note(s): [1] TN, total nitrogen. [2] EC, electrical conductivity.

As shown in Table 2, the leachate characteristics of old landfills varied little in different countries. The value of $BOD_5/COD_{Cr}$ was less than 0.25, and biochemical reactions were not easy to occur. Moreover, the concentration of ammonia nitrogen and salt were at high levels. However, the characteristics of leachate are distinct in different countries due to garbage sources, classification and treatment processes, and so on. The specific reasons are analyzed as follows:

(1) Differences in management regulations lead to differences in original landfill materials. During the earlier years, Europe and the United States moderated the organic matter content of materials entering landfills, and the landfill materials were mainly inorganic substances, while the landfill materials in China included much of the original waste [16]. Colombia and India have not yet carried out waste classification, and Colombian landfills continued to add new waste to the landfills, resulting in high concentrations of $COD_{Cr}$ and $BOD_5$ and poor biochemical properties.

(2) Due to different dietary habits, the content of food waste in Chinese municipal solid waste was much higher than that in the United States, Japan, and Europe. Therefore, the concentrations of $COD_{Cr}$ and $BOD_5$ in Chinese late-stage landfill leachate were higher than those in the United States, Japan, and Europe

(3)   In terms of the scale of landfills, municipal solid waste landfills in China, especially in first-tier cities, had a much larger daily landfill scale and landfill storage capacity than similar landfills in the United States, Japan, and Europe. The anaerobic environment was more thorough, and oxygenated air struggled to enter the landfill; hence, the concentrations of $COD_{Cr}$ and $BOD_5$ were higher in China.

### 2.2. Characteristics and Influencing Factors of Leachate in Incineration Power Plants and Transfer Stations

The construction of waste incineration power plants in China started at the beginning of the 21st century and developed rapidly over the past 20 years. Due to the difference in the generation mechanism, operation technology, generation source, etc., the influencing factors of incineration power plant leachate were quite different from the leachate of the landfill. The leachate of the incineration power plant is mainly produced from the material pit, which is used to adjust the peak amount of garbage and increase the calorific value of the incinerated refuse that builds up in the pit for approximately 7 days [17]. The leachate characteristics are mainly related to the type of garbage collection and transportation system, the composition of the garbage, and the storage period of the material pit. Similar to landfill leachate, leachate from incineration power plants also has the characteristics of a large organic load, complex water quality, high salt content, and large water fluctuation. Its water quality characteristics are shown in Table 3.

**Table 3.** Leachate characteristics from incineration power plants in China.

| Classification | pH | $COD_{Cr}$ (mg/L) | $BOD_5$ (mg/L) | $BOD_5/COD_{Cr}$ | $NH_3$-N (mg/L) | Salt Content (mg/L) |
|---|---|---|---|---|---|---|
| Incineration plant | 6–7 | 60,000–80,000 | 25,000–35,000 | 0.45–0.5 | 1500–2500 | 8000–12,000 |

According to Table 3, compared to landfill leachate, the leachate from incineration power plants, named "fresh" leachate, has a higher pollutant concentration and better biodegradability. The construction period of transfer stations were in the same period as incineration power plants. The water quality and its influencing factors have the same characteristics as the leachate of the waste incineration power plant. In practical applications, the leachate from incineration power plants and transfer stations can be mutually referenced.

### 2.3. Characteristics and Influencing Factors of Anaerobic Digested Slurry of Food Waste

In recent years, waste classification has been promoted, and the classified perishable food wastes were mainly treated through pretreatment + wet anaerobic fermentation or dry anaerobic fermentation processes [18].

Because of the anaerobic system treatment, the leachate characteristics of the dehydrated digested slurry are similar to the landfill leachate in the middle and late stages. However, the pollution concentration as well as the salt content is slightly higher than that of the landfill leachate due to the difference in the production mechanism and operating environment. The characteristics of anaerobic digested slurry from food waste are shown in Table 4.

**Table 4.** Characteristics from anaerobic digested slurry in China.

| Classification | pH | $COD_{Cr}$ (mg/L) | $BOD_5$ (mg/L) | $BOD_5/COD_{Cr}$ | $NH_3$-N (mg/L) | Salt Content (mg/L) |
|---|---|---|---|---|---|---|
| Anaerobic digested slurry | 6–9 | 8000–15,000 | 4000–7000 | - | 2000–3000 | 10,000–20,000 |

### 2.4. Comparison of Three Kinds of Leachate Characteristics

The three kinds of leachate characteristics from different treatments and disposal plants are shown in Table 5. Although the $COD_{Cr}$, $BOD_5$, ammonia nitrogen, and salt contents of the three leachates are generally high, the leachate characteristics from different sources vary due to different production mechanisms, fermentation cycles, and waste

compositions. With the increase in landfill years, the ammonia nitrogen concentration of landfill leachate increases, the biodegradability decreases, and it becomes more difficult to treat the material. Compared with the United States, Japan, and Europe, the pollutant concentration of late-stage landfills is higher. The leachate produced from incineration power plants has a higher pollutant concentration and readily biodegrades due to its original "freshness". Due to the source of cooked food, the food waste anaerobic digested slurry has the highest salt content, but the biochemical properties are similar to the leachate characteristics in the middle and late stages of the landfill.

**Table 5.** Leachate characteristics from three different sources.

| Classification | | pH | $COD_{Cr}$ (mg/L) | $BOD_5$ (mg/L) | $NH_3$-N (mg/L) | Salt Content (mg/L) |
|---|---|---|---|---|---|---|
| Landfill | early | 6.0–6.5 | 10,000–25,000 | 1200–12,000 | 100–1200 | 2000–14,000 |
| | middle | 6.5–7.5 | 6000–12,000 | 1500–4000 | 1000–2000 | 6000–15,000 |
| | late | 7.5–9.0 | 500–7000 | 100–1500 | 100–3000 | 2000–14,000 |
| Incineration plant or transfer station | | 6.0–7.0 | 60,000–80,000 | 25,000–35,000 | 1500–2500 | 8000–12,000 |
| Food waste digestion plant | | 6.0–9.0 | 8000–15,000 | 4000–7000 | 2000–3000 | 10,000–20,000 |

## 3. Development History of the Leachate Treatment Process

Restricted by the level of economic development, the construction of leachate treatment plants in China is relatively delayed. From the perspective of time, leachate characteristics, and treatment processes, the treatment of leachate in China was divided into five stages, as summarized in Table 6.

**Table 6.** Influent and effluent characteristics of leachate treatment from municipal solid waste treatment and disposal plants at different development stages.

| Stage | Main Treatment Process | Water Sample | pH | $COD_{Cr}$ (mg/L) | $BOD_5$ (mg/L) | $NH_3$-N (mg/L) | SS [1] (mg/L) | Cost [3] (CNY/m³) | Typical Case |
|---|---|---|---|---|---|---|---|---|---|
| 1st | hypoxic tank + sedimentation tank + aerobic tank + sedimentation tank | Influent | 6–9 | 6000 | 3000 | - [2] | - | - | Hangzhou Tianziling Landfill |
| | | Effluent | 6–9 | 300 | 60 | - | 100 | | |
| 2nd | ammonia stripping + anaerobic biological filter + SBR reaction tank | Influent | 5.6–7.5 | 3000–13,000 | 1000–26,000 | 400–1500 | - | 4–5 | Shenzhen Xiaping Landfill |
| | | Effluent | 6–9 | 500 | 300 | - | - | | |
| 3rd | two-stage disc tube reverse osmosis (DT-RO) process | Influent | 6–9 | 6000–8000 | 3000–4000 | 1500–3000 | - | 12–15 | Chongqing Chang-shengqiao Landfill |
| | | Effluent | - | 100 | 30 | 15 | - | | |
| 4th | IC anaerobic reactor + two-stage A/O + ultrafiltration + nanofiltration (NF) + reverse osmosis (RO) | Influent | 6.5–7.5 | 75,000 | 30,000 | 2000 | 10,000 | 30–40 | Xuzhou No. 2 Incineration Plant |
| | | Effluent | 6.5–8.5 | 60 | 10 | 10 | 180 | | |
| 5th | pretreatment + two-stage A/O + MBR + Fenton advanced oxidation + BAF | Influent | 7.5–8.5 | 15,000–18,000 | 6000–8000 | 2800–3000 | 2000–2500 | 40–75 | Hefei Xiaomiao Food Waste Resource Center |
| | | Effluent | 6.0–8.0 | 300 | 150 | 35 | 200 | | |

Note(s): [1] SS, suspended solids. [2] -, not detected. [3], China yuan.

### 3.1. The First Stage: Simple Biological Treatment during Early 1990s

During this stage, the leachate was mainly landfill leachate, and the treatment technology used for this stage in China was vacuum treatment. The treatment process mainly refers to the treatment method of urban sewage and adopts the method of biological treatment. The representative project is the Hangzhou Tianziling First Landfill Leachate Treatment Plan. The typical influent and effluent characteristics of the first stage from the Hangzhou Tianziling landfill are shown in Table 6.

The Hangzhou Tianziling First Landfill Leachate Treatment Plan adopted the two-stage activated sludge method, using the main process of "hypoxic tank + sedimentation tank + aerobic tank + sedimentation tank". This treatment effect was good for its time. In the first step of treatment development, the mixture of bacteria and low-grade molds is the dominant strain. In the second step, the cultivated protozoa are the dominant strain. The two steps have different requirements for the concentration control of DO and MLSS. To reduce operating costs, the effect of two-step treatment can achieve the standard emission requirements. The disadvantage is that the "low oxygen-aerobic" influent concentration requires $COD_{Cr}$ to be approximately 6000 mg/L, which is not suitable for high-concentration organic wastewater [19].

At this stage, since the construction of the landfill leachate treatment plant mainly refers to the urban sewage treatment plant, the water quality characteristics of the leachate are not considered, and there is no reference to the research and practice results in this area [20].

### 3.2. The Second Stage: Ammonia Stripping + Anaerobic + Aerobic Treatment during the Mid to Late 1990s

At this stage, researchers considered the unique water quality of landfill leachate, such as high concentrations of organic matter and ammonia nitrogen. This research resulted in deamination measures being adopted [21]. The treatment process is generally a combination of ammonia stripping + anaerobic + aerobic treatment, and the effluent was directly discharged into the urban sewers after treatment. Representative projects for this treatment include the Jiangmen Datsuche Mountain and Shenzhen Xiaping Landfill leachate treatment stations. Table 6 shows the leachate characteristics from the Shenzhen Xiaping Landfill as a reference to represent the characteristics of the treatment effect at this stage.

The more specific application process is "ammonia stripping + anaerobic biological filter + SBR reaction tank", and the effluent meets the third-level standard of the "Comprehensive Wastewater Discharge Standard" (GB8978-1996). This process adopts a chemically structured packed tower, and $NH_3$-N is removed from the leachate under precise control of pH adjustment [22]. This technology effectively solves the problem of deamination of the leachate, and the ammonia nitrogen in the effluent remains at approximately 10 mg/L. At the same time, the C/N nutrient ratio balance in the leachate should be controlled to ensure that the subsequent anaerobic and aerobic biological treatment has a high removal efficiency.

At this stage, Shenzhen realized that the components of urban landfill waste contain a large amount of food waste and exclude coal ash. Therefore, the leachate has high concentrations of organic matter and ammonia nitrogen. The leachate composition of this kind of landfill is quite distinct from the leachate containing a lot of fly ash landfill. Different leachate treatment methods were applied to adapt to the differences in leachate characteristics [23]. Treatment technology at this stage was similar to that used in Brazil, where uses biological technology even now [24].

### 3.3. The Third Stage: Two-Stage Disc Tube Reverse Osmosis (DT-RO) Process from 2000 to 2008

Due to rapid economic development, new landfill leachate treatment plants are generally far from urban areas, and the leachate has no route to be discharged into the urban sewage pipe network. Therefore, the treatment requirements are also increased accordingly and generally need to be treated to satisfy the secondary or even primary discharge standards of the "Comprehensive Wastewater Discharge Standard" (GB8978-1996) and the "Landfill Pollution Control Standard for Domestic Waste" (GB 16889-1997). The first-tier and second-tier cities have adopted the biological treatment process. However, at this time, the leachate characteristics of some large-scale landfills have also changed greatly due to factors such as prolonged landfill life, and the expected requirements of treatment cannot be met only by biological treatment. The representative projects at this stage include the Guangzhou Xingfeng and Chongqing Changshengqiao Landfill Leachate Treatment Plants. The treatment effect of the Chongqing Changshengqiao Landfill Project is shown in Table 6.

The core treatment technology adopts the two-stage disc tube reverse osmosis (DT-RO) process, and the effluent meets the first-class standard of the "Landfill Pollution Control Standard for Domestic Waste" (GB 16889-1997) [25,26]. The concentrate of the first-stage reverse osmosis is discharged to the landfill stack, and the concentrate of the second-stage reverse osmosis is recharged to the water inlet of the sand filter. The application of membrane technology for landfill leachate treatment was adopted around 10 years later than in other developed countries. A previous study reported that membrane-based leachate treatment technologies had been applied in landfills of some European countries, such as Germany [27].

The water quality of the early effluent met the discharge requirements and solved the problem of zero discharge of pollutants at that time. However, with the increase in treatment years, the accumulation of concentrated liquid in the landfill pile became increasingly serious. At this time, some developed cities in China had built waste incineration power plants and small- and medium-sized transfer stations. The leachate of the waste incineration power plants and transfer stations mainly adopts a pretreatment + anaerobic + aerobic + advanced treatment process. The effluent quality reached the "Comprehensive Wastewater Discharge Standard" GB8978 requirements for external discharge into municipal sewage pipe networks. In addition, environmental protection requirements were relaxed during this period, and many small- and medium-sized transfer stations produced a small amount of leachate and were directly transported to sewage treatment plants without treatment at transfer stations.

### 3.4. The Fourth Stage: Anaerobic + Aerobic + Advanced Treatment Process from 2008 to 2015

The fourth stage is after the "Landfill Pollution Control Standard for Domestic Waste" (GB16889-2008) was promulgated. This standard requires that $COD_{Cr}$ is below 100 mg/L and $NH_3$-N is below 25 mg/L. Due to the increasingly stringent emission standards, traditional biological treatment processes cannot meet the requirements of the new standards [28], so the major landfills across the country have made technological improvements, which utilize membrane treatment technology [29]. The treatment ability of landfill leachate has been greatly improved, but the existing problems are as follows:

(1)　The operating cost is high. The external MBR system used in the traditional process system consumes a large amount of electricity [30], and the operating cost is high [31]. In addition, due to the imbalance of the C/N ratio in the aged landfill, a large amount of the carbon source needs to be supplemented, resulting in a higher operating cost.

(2)　To ensure that the final treated effluent satisfies the standard requirements, the advanced treatment adopts the NF + RO process [32]. The membrane treatment process inevitably produces a large amount of intractable concentrate. At present, the material of NF concentrate through the material membrane can be solved by ozone catalytic oxidation or other processes, but the operation cost is relatively high. For RO concentrate, there is no mature and stable treatment process in China, and the concentrate produced by the reverse osmosis system contains a large amount of salt. The untreated concentrate is directly returned to the leachate treatment system, which will reduce the activity of microorganisms. This operation leads to serious membrane fouling and accelerates the frequency of membrane cleaning. Therefore, the life of the membrane is reduced [33]. If the untreated concentrate is directly sprayed back to the landfill stack, the salt will accumulate in the landfill. This technology significantly affects the normal operation of the landfill leachate treatment system. Therefore, the concentrate produced by the membrane system must be treated.

(3)　The problem of concentrate caused by membrane treatment technology has gradually attracted increasing attention. Enterprises and universities have begun to focus on the development of advanced oxidation and evaporation treatment technology. Evaporation processes include MVR and immersion combustion. The evaporation process also has a series of problems, such as high energy consumption and scaling difficulties when cleaning, and the stability and reliability of operation need to be further verified.

At this stage, Chinese waste incineration power plants and waste transfer stations entered rapid development. The leachate of waste incineration power plants mainly adopts a pretreatment + anaerobic + aerobic + advanced treatment process, and the effluent satisfies the recycled water standard and is reused at incineration plants to save water. The leachate of waste transfer stations is treated to meet the "Landfill Pollution Control Standard for Domestic Waste" (GB16889-2008) or "Comprehensive Wastewater Discharge Standard" (GB8978-1996) and discharged into the municipal sewer network. The treatment effect of typical cases, such as the Xuzhou No. 2 Incineration Plant, is shown in Table 6.

The leachate treatment process of the Xuzhou No. 2 Incineration Plant is "IC anaerobic reactor + two-stage A/O + ultrafiltration + nanofiltration (NF) + reverse osmosis (RO)". The designed effluent quality reaches the open circulating cooling water system supplementary water quality standards in "Urban Wastewater Recycling and Utilization Industrial Water Quality" (GB/T19923-2005). The $COD_{Cr}$ in the leachate is as high as 75,000 mg/L, and a high-efficiency anaerobic reactor needs to be set up to reduce the $COD_{Cr}$ to approximately 15,000 mg/L. The biochemical system adopts the two-stage A/O + UF process that is currently stable and reliable in the domestic leachate treatment industry. The two-stage A/O + UF process has strong resistance to hydraulic and water quality impact load capacity. A deep denitrification reaction is used to ensure biochemical denitrification efficiency. Advanced treatment adopts the NF + RO process to remove refractory organic matter and ensure that the effluent TDS meets the requirements of water quality standards for reuse.

To collaboratively solve the energy demand and byproducts of leachate treatment, the advantages of incineration plants are fully utilized. If the effluent meets the standard requirements of water quality for reuse, it can be reused as circulating cooling recharge water in the factory area. The concentrates containing a high concentration of organic matter adopt a back injection incinerator scheme, which effectively solves the problem of concentrated liquid, and the current operation is in good condition.

### 3.5. The Fifth Stage: Diversified and Full Quantitative Process since 2015

In the fifth stage, the waste terminal treatment pattern gradually entered the period of "incineration as the mainstay and landfill to support the bottom" in China [34].

Incineration power plants can consume concentrated liquid in combination with their own water use characteristics. Therefore, the leachate produced by the incineration power plant is still required to meet the water reuse standard, and the advanced treatment is still the nanofiltration + reverse osmosis combined process or high-pressure reverse osmosis process. The mainstream process of landfill leachate still adopts biological treatment + advanced treatment processes. During this period, the aged leachate treatment industry focused on denitrification pretreatment and concentrate treatment. Denitrification pretreatment mainly adopts anaerobic ammonia oxidation, multistage A/O, and stripping ammonia removal processes. Concentrate treatment mainly adopts immersion combustion evaporation, low-temperature negative pressure evaporation, and an improved mechanical vapor recompression (MVR) process.

With the active promotion of waste classification, the treatment of food waste digested slurry has become an area of focus in the development of the industry. The treatment process refers to the landfill leachate treatment process. However, the digested slurry contains many impurities, and reasonable selection of the pretreatment process has a high influence on the subsequent operation. The treatment effect of typical cases, such as the Hefei Xiaomiao Organic Waste Resource Processing Center, is shown in Table 6.

The digested slurry treatment adopts the combined treatment process of "pretreatment + two-stage A/O + MBR + Fenton advanced oxidation + BAF" [35], and the treated effluent meets the takeover standard of the local sewage treatment plant. The process adopts an efficient pretreatment system using an oil separation + air flotation + membrane fine grid combination process to efficiently remove pollutants such as suspended solids, TP, and oil. The biochemical system adopts a two-stage A/O + ultrafiltration process, which is a stable and reliable treatment process in China. The two-stage A/O + ultrafiltration process has the

characteristics of strong impact load resistance and good removal of organic matter and nitrogen. Advanced treatment adopts a Fenton advanced oxidation + aerated biological filter combined process to remove refractory organic matter and ensure that effluent $COD_{Cr}$ and TN meet emission standards [36]. This process does not have the problem of concentrating produced by traditional membrane methods and currently operates in good condition, but the processing cost is high. Currently, the advanced oxidation process is an intensive method for mature leachate treatment. Furthermore, the resulting products generated by the advanced oxidation process are assimilated by microorganisms in the biological treatment process [37]. Thus, the advanced oxidation process combined with biological technology is gradually becoming widely used in the field of leachate treatment [38].

According to incomplete statistics, the current proportion of the main leachate treatment technologies in China is shown in Figure 1. The proportion of MBR + membrane advanced treatment was 78%, occupying the main application position. With the continuous development of science and technology, the leachate treatment process will show a trend of diversification and full quantification. With the deepening of the understanding of leachate treatment, the selection of leachate treatment processes will be more rigorous, scientific, and reasonable and can achieve ecological and environmental protection and low carbon emissions.

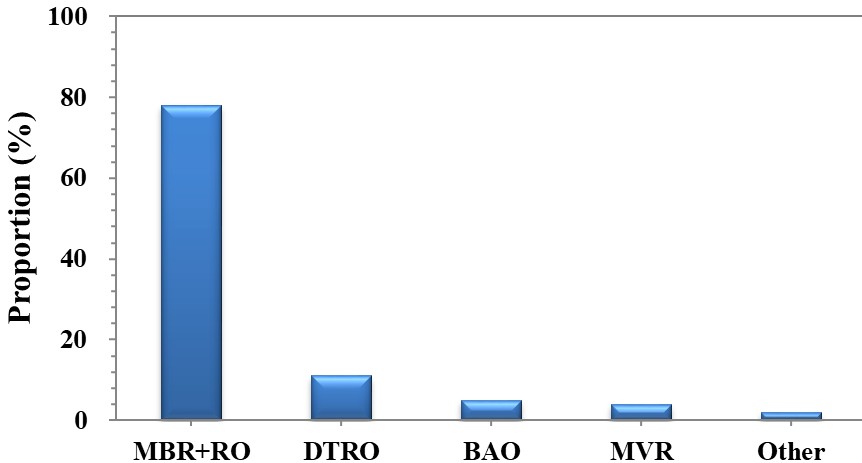

**Figure 1.** The proportion of main leachate treatment technologies. MBR + RO: membrane bioreactor + reverse osmosis; DTRO: two-stage disc tube reverse osmosis; BAO: biochemical + advanced oxidation; MVR: mechanical vapor recompression.

*3.6. Economic Analysis of Treatment Technologies*

With the development of science and technology and the gradual stricter discharge standards, the treatment and disposal technology of landfill leachate has been continuously improved. The processing costs of landfill leachate in different stages are summarized in Table 6, which shows that the cost increased continuously from 4 to 75 CNY/m$^3$. As mentioned above, technology of landfill leachate treatment was gradually changed from physical technology to chemical–physical technology. Recently, membrane technology has gradually been incorporated in landfill leachate technology, which was the main reason for the increase in treatment cost [39]. Therefore, there is an urgent need to find cost-effective technologies to treat landfill leachate to meet increasingly stringent discharge requirements.

**4. Future Development Trends**

The landfill leachate industry and even the entire environmental protection industry serve the national economy, people's livelihoods, and social and economic development; thus, its future development must meet the needs of the Chinese social economy and urban development. From the perspective of the leachate industry, its future development trends are mainly illustrated in Figure 2.

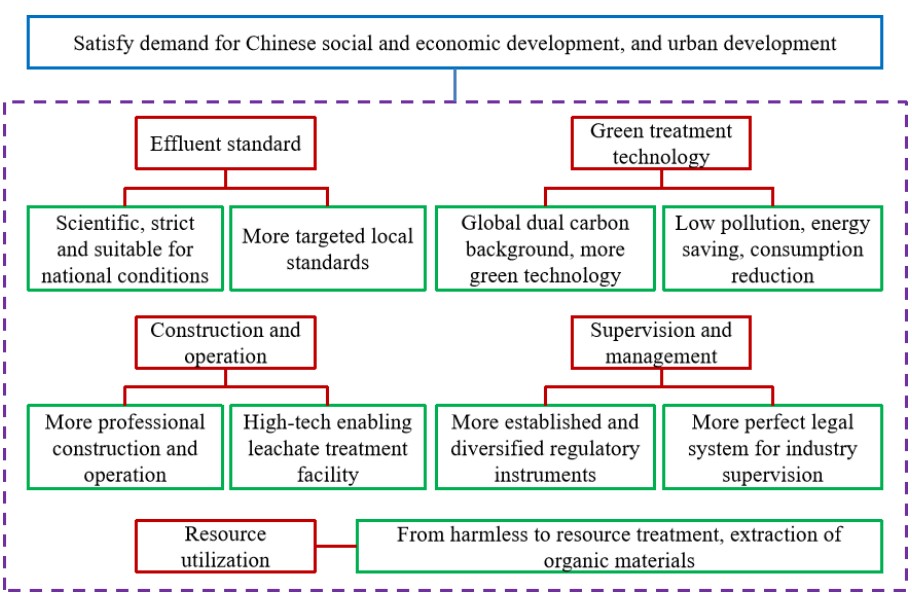

**Figure 2.** Thumbnail of future leachate development tendency.

*4.1. Increased Scientific Emission Standards and Tendency of Promulgating Local Standards*

The discharge standards for effluent of Chinese waste leachate vary according to different discharge routes and uses, but the discharge standards for leachate treatment pollutants in some areas lack scientific significance. Therefore, it is urgent to establish higher scientific and reasonable national and local discharge standards for leachate treatments that are suitable for Chinese national conditions [40]. In the long run, to create a good ecological environment, it is the overall trend that pollutant discharge standards in various industries are more scientific. The field of leachate treatment is no exception, and future discharge standards will be more scientific. Taking the leachate treatment of a municipal solid waste incineration power plant as an example, the leachate is treated in the factory and then discharged into the urban sewage treatment plant. Generally, the effluent generally needs to comply with the "Water Quality Standard for Sewage Discharged into Urban Sewers" (GB/T 31962-2015) or the third-level standard in "Comprehensive Wastewater Discharge Standard" (GB 8978-1996), and standards requires $COD_{Cr} < 500$ mg/L. In fact, for leachate after biochemical treatment in incineration plants, the effluent can meet the requirement of $COD_{Cr} < 500$ mg/L, although $BOD_5$ is almost exhausted, and it is difficult to treat after being discharged into municipal sewage treatment plants [41]. These synergistic factors will be considered in future standard development.

There are many subdivisions of leachate in China, and the economic development level, regional environment, supporting facilities, and local environmental capacity of different regions are quite distinct. However, the leachate treatment discharge standards of different regions execute the same national or industrial standards, and there are no local standards at present. In the future, leachate treatment discharge standards will break the existing pattern of using the same standard in different areas. Similar to the current urban sewage treatment plant, local standards for leachate treatment discharge will be issued according to the conditions of different regions, which are highly targeted, science based, and reasonable.

*4.2. Inevitable Green Development of Processing Technology Route*

In the background of the world's efforts to achieve carbon peaking and carbon neutrality, China, as a major carbon emitter, has put forward a dual-carbon strategy, the goal of which is to achieve carbon peaking by 2030 and carbon neutrality by 2060. To transform Chinese economic development, a low-carbon and green development path was set. China will comprehensively promote the reform of the energy structure, industrial structure,

and consumption structure. Hence, the green development route of leachate treatment technology is the only way in the future.

At present, it is difficult to achieve the standard discharge of $COD_{Cr}$ and total nitrogen in leachate by traditional biological methods. Nanofiltration and reverse osmosis processes have almost become the standard for leachate treatment [42]. The operating cost of leachate treatment is directly increased by membrane service life and power consumption. Considering the operating cost of leachate treatment, in view of the characteristics of high ammonia nitrogen and high organic pollutants in leachate, the future green development trend of treatment technology may be a new high-efficiency denitrification process (such as anammox, short-range digestion and reverse digestion, endogenous denitrification) [43–45], new advanced catalytic oxidation, evaporation, and other fully quantitative process technologies, which can achieve high treatment efficiency, strong adaptability, stable effluent, indicator-satisfying standards, no secondary pollution, fewer auxiliary products (such as the problem of concentrate and sludge), energy savings, consumption reduction, economic rationality, and relatively low operating costs.

### 4.3. Increased Professional and Intelligent Construction and Operation

The current leachate treatment is basically the mode of purchasing services, and most of them are investment enterprises. The lack of professional and technical capabilities in operations management results in idle and wasteful facilities. With the continuous improvement of environmental protection requirements and the continuous deepening of industry segmentation, the construction and operation of leachate treatment facilities will become more and more professional in the future.

With the popularization of big data and the proposal of intelligent cities in China, advanced technology will promote the progress of leachate treatment technology in the future. The main process parameter control system, security system, energy system, and environmental protection system of the entire project will achieve all-around intelligence.

### 4.4. Stricter Supervision and Management

With the increasingly strict standards system and the application of intelligence, the means of supervision of leachate treatment facilities in the future will be more ideal and diversified, eliminating the regulatory loopholes in hardware.

With the development of the Chinese social economy and the continuous improvement of the legal system, the supervision and legal system of the leachate treatment industry will be more complete in the future. The more complete legal system will provide a guarantee for strict supervision and management in terms of software.

### 4.5. Trends in Technology

As mentioned above, with the development of the economy and the continuous stricter emission standards, technologies of landfill leachate become more diversified. Twenty published papers about landfill leachate treatment technologies from the years 2020 to 2022 are summarized in Table 7. It can be seen that physical and chemical technologies were the most widely used in landfill leachate, and biological technology was less used. The reason for this phenomenon was that the landfill leachate generally contained a large amount of refractory organic matter, which was difficult to be degraded by microorganisms. Moreover, more and more researchers paid attention to composite processing technology, such as biological combined with chemical and physical combined with chemical technology. The research trend showed that combining multiple technologies could effectively achieve landfill leachate disposal and reduce environmental pollution.

In addition, current leachate treatment is mainly based on harmless treatment, but because the leachate contains high organic matter, ammonia nitrogen, and salt content, and especially a large amount of humic acid, researchers try to recycle it, such as recycling ammonia nitrogen to make ammonium sulfate, extraction of humic acid fertilizer, anaerobic production of biogas, production of hydrogen, etc., but the leachate recycling technology is

not yet mature and has not been widely applied [46,47]. With the continuous development of technology and the continuous improvement of resource utilization requirements, the path of leachate resource utilization will be diversified and practically applied.

**Table 7.** Treatment technologies of landfill leachate in the years 2020–2022.

| Number | Reference | Treatment Technology | Classification |
|---|---|---|---|
| 1 | [48] | Membrane | Physical |
| 2 | [49] | Adsorption | Physical |
| 3 | [50] | Membrane | Physical |
| 4 | [51] | Nanofiltration | Physical |
| 5 | [52] | Adsorption | Physical |
| 6 | [53] | Membrane | Physical |
| 7 | [54] | Advanced oxidation | Chemical |
| 8 | [55] | Fenton based advanced oxidation | Chemical |
| 9 | [56] | Advanced oxidation | Chemical |
| 10 | [57] | Photocatalytic | Chemical |
| 11 | [58] | Electrocatalytic ozonation | Chemical |
| 12 | [59] | Simultaneous ammonium oxidation denitrifying | Biological |
| 13 | [60] | Advanced oxidation + Denitrification | Biological + Chemical |
| 14 | [61] | Ozonization + microalgae | Biological + Chemical |
| 15 | [62] | Electrocoagulation | Physical + Chemical |
| 16 | [63] | Advanced oxidation +adsorption | Physical + Chemical |
| 17 | [64] | Ion exchange+ supercritical water oxidation | Physical + Chemical |
| 18 | [65] | Ozone direct oxidation + Catalytic oxidation + Membrane | Physical + Chemical |
| 19 | [66] | Coagulation–flocculation | Physical + Chemical |
| 20 | [67] | Coagulation + advanced oxidation | Physical + Chemical |

## 5. Conclusions

(1) The characteristics of the leachate are related to the production mechanism, fermentation cycle, and waste composition. With the increase in landfill years, the concentration of ammonia nitrogen in the landfill leachate increases, the biodegradability decreases, and the C/N is out of balance, resulting in leachate being more difficult to treat. Compared with other countries, although the quality of landfill leachate was similar in the middle and late stages, the concentration of pollutants in the late stage leachate in China was higher, which was attributed to the differences in eating habits, landfill size, and treatment methods.

(2) The leachate treatment process mainly went through five stages. The treatment process and the emission standards have been continuously improving. In recent years, due to the improvement of environmental protection standards and national environmental protection policies, the treatment process of landfill leachate was mainly "secondary A/O + ultrafiltration + nanofiltration (NF) + reverse osmosis (RO)" or "pretreatment + secondary A/O + MBR + Fenton Advanced Oxidation + BAF".

(3) The future development of the landfill leachate treatment industry and the entire environmental protection industry need to meet the requirements of the Chinese social economy and urban development. Under the background of the global "dual carbon" strategy, the green ecology of the technical route is represented by new high-efficiency denitrification, new advanced catalytic oxidation, evaporation, and other fully quantitative process technologies.

**Author Contributions:** Conceptualization, X.S., M.J. and L.Z.; investigation, H.M. and Q.F.; resources, X.D. and X.W.; writing—original draft preparation, X.S.; writing—review and editing, L.L.; visualization, L.L.; supervision, W.Z. All authors have read and agreed to the published version of the manuscript.

**Funding:** This research was funded by the Key Technologies R&D Program of Tianjin, China, grant number 19ZXSZSN00080.

**Conflicts of Interest:** The authors declare no conflict of interest.

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
