# Peer review of "The Experience and Development of the Treatment Technology of Municipal Solid Waste Leachate in China"

_water, doi:10.3390/w14162458_

Round 1

Reviewer 1 Report

 The paper “The experience and development of the treatment technology of municipal solid waste leachate in China” is of interest to the scientific community. However, major revisions are required for improving the paper

 Specific comments:

-line 17 -  the last paragraph of the Abstact - “Furthermore, under the requirements of ecological environmental protection and “Dual carbon” development concepts, this paper advised the future development trend of the Chinese municipal solid waste leachate, which will enact more scientific emission standards and local standards, become inevitable for the green ecology of the technical route, be more professional and intelligent for the construction and operation, and diversify resource utilization of the treatment facilities process.” Is too long and confusing. Thus, it must be simplified using short phrases as concise ideas.

-line 26 – 63 - Introduction – This section needs a substantial improvement. It is not possible to support a review paper using only one reference in this section. Thus, not only more references need to be cited in this section, but also some phrases and ideas need to be polished.

- line 59, the last paragraph needs improvement in order to clarify the objective.

- line 184, Table 5 must be Table 6.

- The paper will benefit significantly if a section with costs (e.g. cost/cubic meter treated) is included. This information may be summarized in a Table.

-line 454 – Conclusions – The division in 3 points is all right, but the length of this section should be significantly reduced, perhaps to half the size.

Author Response

Comment (1): line 17 - the last paragraph of the Abstact - “Furthermore, under the requirements of ecological environmental protection and “Dual carbon” development concepts, this paper advised the future development trend of the Chinese municipal solid waste leachate, which will enact more scientific emission standards and local standards, become inevitable for the green ecology of the technical route, be more professional and intelligent for the construction and operation, and diversify resource utilization of the treatment facilities process.” Is too long and confusing. Thus, it must be simplified using short phrases as concise ideas.

Response: Thanks for the reviewer’s suggestion. We rewrote this part. Please see it as follows.

Furthermore, under the requirements of ecological environmental protection and the development concept of "Dual Carbon", this paper provided suggestions on the treatment and disposal technologies of urban solid waste leachate in China in the future, making the treatment process more professional and intelligent, which is conducive to resource utilization.

Comment (2): line 26 – 63 - Introduction – This section needs a substantial improvement. It is not possible to support a review paper using only one reference in this section. Thus, not only more references need to be cited in this section, but also some phrases and ideas need to be polished.

Response: We really appreciate the suggestion of reviewer’s. We rewrote the Introduction. Please see it as follows.

The process of development of the leachate treatment industry corresponded to the development of the Chinese waste treatment industry during different periods. Originally, municipal solid waste disposal in China could be dates back to the 1980s, and the disposal method gradually developed from simple stacking of waste to the modern sanitary landfills[1]. Leachate is a high-concentration organic wastewater which is produced in the process of garbage collection, transportation and disposal[2]. Generally, leachate was mainly divided into landfill leachate, incineration plant, transfer station leachate, and also organic waste anaerobic digested slurry.

With the continuous improvement of the landfill leachate collection system and environmental protection regulations in China, landfill leachate had become a key pollutant in sanitary landfills[3]. These regulations had driven the development of Chinese landfill leachate treatment industry. With the rapid development of economy in China, various waste treatment and disposal technologies also developed rapidly. For example, waste incineration power plants gradually replaced sanitary landfills as a new mainstream waste treatment method[4]. In addition, due to the rapid development of urbanization and increase of garbage collection and transportation distance, the application of transfer stations had gradually been promoted. Since the high-concentration wastewater was collected by waste incineration power plants and transfer stations had similar properties to fresh landfill leachate, these three kinds of wastewater are collectively referred to as leachate. In 2010, four ministries and commissions, including the National Development and Reform Commission and the Ministry of Agriculture of China, issued a notice to select qualified cities or municipalities to carry out pilot work on resource utilization and harmless disposal of kitchen waste. In 2016, China began to generally implement waste classification nationwide[5]. By the end of 2021, almost cities in China with a population of more than one million had built kitchen waste recycling facilities [6]. Household food waste is generated during the waste classification process, and its physical composition is similar to that of restaurant kitchen waste. These two types of waste are collectively referred to as food waste. The mainstream process technology route adopted for food waste is the pretreatment + wet or dry anaerobic fermentation process[7]. The digested slurry had similar properties to landfill leachate, and it was also classified into municipal solid waste leachate[8].

Due to the complexity and uncertainty of landfill leachate, the disposal technology of landfill leachate was very complicated. In order to improve the ecological environment and provide cost-effective treatment options for managers, the objectives of this review are (1) to summarize the historical process of municipal solid waste leachate treatment in China and analyze the characteristics of landfill in different countries, (2) to summarize processes of leachate treatment and discharge standards at different stages in China, (3) and to prospect and advise the technology for leachate treatment and standards establishment for the future.

References:

  1. Karak, T.; Bhagat, R.; Bhattacharyya, P. Municipal solid waste generation, composition, and management: the world scenario. Critical Reviews in Environmental Science and Technology 2012, 42, 1509-1630.
  2. Luo, H.; Zeng, Y.; Cheng, Y.; He, D.; Pan, X. Recent advances in municipal landfill leachate: A review focusing on its characteristics, treatment, and toxicity assessment. Science of the Total Environment 2020, 703, 135468.
  3. Rajoo, K.S.; Karam, D.S.; Ismail, A.; Arifin, A. Evaluating the leachate contamination impact of landfills and open dumpsites from developing countries using the proposed Leachate Pollution Index for Developing Countries (LPIDC). Environmental Nanotechnology, Monitoring & Management 2020, 14, 100372.
  4. Li, Y.; Zhao, X.; Li, Y.; Li, X. Waste incineration industry and development policies in China. Waste Management 2015, 46, 234-241.
  5. Zhou, M.-H.; Shen, S.-L.; Xu, Y.-S.; Zhou, A.-N. New policy and implementation of municipal solid waste classification in Shanghai, China. International journal of environmental research and public health 2019, 16, 3099.
  6. De Clercq, D.; Wen, Z.; Fan, F.; Caicedo, L. Biomethane production potential from restaurant food waste in megacities and project level-bottlenecks: a case study in Beijing. Renewable and Sustainable Energy Reviews 2016, 59, 1676-1685.
  7. Angelonidi, E.; Smith, S.R. A comparison of wet and dry anaerobic digestion processes for the treatment of municipal solid waste and food waste. Water and environment journal 2015, 29, 549-557.
  8. Jayanth, T.; Mamindlapelli, N.K.; Begum, S.; Arelli, V.; Juntupally, S.; Ahuja, S.; Dugyala, S.K.; Anupoju, G.R. Anaerobic mono and co-digestion of organic fraction of municipal solid waste and landfill leachate at industrial scale: Impact of volatile organic loading rate on reaction kinetics, biogas yield and microbial diversity. Science of The Total Environment 2020, 748, 142462.

Comment (3): line 59, the last paragraph needs improvement in order to clarify the objective.

Response: Thanks for the reviewer’s suggestion. We rewrote this paragraph. Please see it as follows.

Due to the complexity and uncertainty of landfill leachate, the disposal technology of landfill leachate was very complicated. In order to improve the ecological environment and provide cost-effective treatment options for managers, the objectives of this review are (1) to summarize the historical process of municipal solid waste leachate treatment in China and analyze the characteristics of landfill in different countries, (2) to summarize processes of leachate treatment and discharge standards at different stages in China, (3) and to prospect and advise the technology for leachate treatment and standards establishment for the future.

Comment (4): line 184, Table 5 must be Table 6.

Response: Thanks for the reviewer’s advice. We appologized for this mistake. We corrected this error. Please see it as follows.

Table 6. Influent and effluent characteristics of leachate treatment from municipal solid waste treatment and disposal plants at different development stages.

Comment (5): The paper will benefit significantly if a section with costs (e.g. cost/cubic meter treated) is included. This information may be summarized in a Table.

Response: Thanks for the reviewer’s suggestion. We supplemented the cost statistics in Table 6. Please see it as follows.

Table 6. Influent and effluent characteristics of leachate treatment from municipal solid waste treatment and disposal plants at different development stages.

Stage

Main treatment process

Water sample

pH

CODCr

(mg/L)

BOD5

(mg/L)

NH3-N

(mg/L)

SS1

(mg/L)

Cost

(RMB/m3)

Typical case

1st

hypoxic tank + sedimentation tank + aerobic tank + sedimentation tank

Influent

6~9

6000

3000

-b

-

-

Hangzhou Tianziling Landfill

Effluent

6~9

300

60

-

100

2nd

ammonia stripping + anaerobic biological filter + SBR reaction tank

Influent

5.6~7.5

3000~

13000

1000~26000

400~

1500

-

4~5

Shenzhen Xiaping Landfill

Effluent

6~9

500

300

-

-

3rd

two-stage disc tube reverse osmosis (DT-RO) process

Influent

6~9

6000~

8000

3000~4000

1500~

3000

-

12~15

ChongqingChangshengqiao Landfill

Effluent

-

100

30

15

-

4th

IC anaerobic reactor + two-stage A/O + ultrafiltration + nanofiltration (NF) + reverse osmosis (RO)

Influent

6.5~7.5

75000

30000

2000

10000

30~40

Xuzhou No. 2 Incineration Plant

Effluent

6.5~8.5

60

10

10

180

5th

pretreatment + two-stage A/O + MBR + Fenton advanced oxidation + BAF

Influent

7.5~8.5

15000~

18000

6000~8000

2800~

3000

2000~

2500

40~75

Hefei Xiaomiao Food Waste Resource Center

Effluent

6.0~8.0

300

150

35

200

Comment (6): line 454 – Conclusions – The division in 3 points is all right, but the length of this section should be significantly reduced, perhaps to half the size.

Response: Thanks for the reviewer’s advice. We rewrote the conclusion. Please see it as follows.

(1) The characteristics of the leachate are related to the production mechanism, fermentation cycle and waste composition. With the increase in landfill years, the concentration of ammonia nitrogen in the landfill leachate increases, the biodegradability decreases, and the C/N is out of balance, resulting in leachate being more difficult to treat. Compared with other countries, although the quality of landfill leachate was similar in the middle and late stages, the concentration of pollutants in the late stage leachate in China was higher, which was attributed to the differences in eating habits, landfill size and treatment methods. The leachate generated from incineration power plants had high biochemical properties and pollutant concentrations because it originated from fresh leachate. The leachate from food waste anaerobic digestion had the highest salt content due to the source of cooking additives.

(2) The leachate treatment process mainly underwent through five stages. The treatment process and the emission standard has been continuously improved. In recent years, due to the improvement of environmental protection standards and national environmental protection policies, the treatment process of landfill leachate was mainly “secondary A/O + ultrafiltration + nanofiltration (NF) + reverse osmosis (RO)” or “pretreatment + secondary A/O+MBR+Fenton Advanced Oxidation+BAF”.

(3) The future development of the landfill leachate treatment industry and the entire environmental protection industry need to meet the requirements of the Chinese social economy and urban development. Under the current general conditions, the future development trend of the leachate treatment industry should enact more scientific emission standards and local standards. Under the background of the global " Dual carbon " strategy, the green ecology of the technical route is represented by new high-efficiency denitrification, new advanced catalytic oxidation, evaporation and other fully quantitative process technologies. In addition, the construction and operation of landfills need to be more professional and intelligent, and the government also needs to formulate strict emission standards.

Reviewer 2 Report

Manuscript ID: water-1799138

The experience and development of the treatment technology of municipal solid waste leachate in China

In this review manuscript, the authors presented the development of the treatment technology of leachate and compared the details with other countries around the world. Also, propose a few suggestions for the future.  

The first 5 pages or almost half of the manuscript deal with the background of the leachate definition, characteristics, and collection.  Only in the second half, the topic as stated in the title is presented. I would suggest shortening the earlier part and focusing more on the treatment technology as suggested by the title.

Section 1: no references are included to support (only 1 reference included)

Most of the tables presented lack the source from where the data presented are obtained.

Line 96: stated that “in China have basically achieved a “zero landfill”, ..” could be nice if the authors could include how it was achieved.

Section 2.4: repetition of fact. no new point is discussed in this comparison.

Figure 1: repetition of section 1.

Throughout the manuscript Table 5 is referred to as Table 6.

It would be nice if the authors could include an economic analysis and sustainability analysis of the 5 stages of the Development history of leachate treatment.  (3.1-3.5). Now it is just the chronology of the development of the 5 stages without depth analysis.

Section 4.5: this section need to be enhanced with more input and details.

Conclusion: this section is lengthy and is just a repetition of facts in previous sections, could avoid repeating points presented earlier. 

Reviewer 3 Report

Dear Authors

Your article addresses interesting topics for a better understanding of the experience of treatment technology of municipal solid waste leachate in China, a fundamental aspect for selecting the most appropriate treatment in order to minimize the environmental impacts that are potentially associated with it when it is discharged. However, I have a few comments/suggestions:

1)      You should correct the numbering of tables and figures: for example, on page 6 you refer Table 5, which I think is Table 6 that you refer on page 7, section 3.1, 1st paragraph. I also suggest that this table be inserted after this paragraph.

2)      In section 4, in my opinion in the text you should refer to Figure 3 (which is also badly numbered as it appears as Figure 2). I am also of the opinion that this entire section should be revised, as it is not clear to me the justification for the future development of what you have called the leachate industry and which you have proposed in the aforementioned figure.

3)      When comparing the characteristics of the leachate from the landfills you studied with the leachate from landfills in other parts of the world (Table 2), I think it would have been interesting to provide some information on the average age of these landfills and on the characteristics of the municipal waste deposited in these countries.

4)      In section 2.2 it was not clear to me which factors explain the higher pollutant organic concentration and better biodegradability of leachate from incineration power plants.

5)      Regarding the results presented in Table 6 (section 3), I would have liked to have seen some discussion/comparison with results obtained by other authors who studied systems with the same type of treatment.

Round 2

Reviewer 1 Report

The comments have been addressed so the article can be published.

Author Response

We greatly appreciate the help of the reviewers. Best wishes.

Reviewer 2 Report

Manuscript ID Water-1799138

The experience and development of the treatment technology of municipal solid waste leachate in China

The authors have carried out the amendments suggested by the reviewer. However, there are a few more comments:

Line 63: “…analyze the characteristics of landfill in different countries..” stated that this is one of the objectives of this manuscript. I would suggest to avoid this statement, as the manuscript did not deal this in depth.

Line 185: Table 6 appears before the text. Would suggest to include the table after the text.  

Conclusion: avoid numbering the points is the conclusion. Would suggest the length of the conclusion.

Author Response

Comment (1): Line 63: “…analyze the characteristics of landfill in different countries..” stated that this is one of the objectives of this manuscript. I would suggest to avoid this statement, as the manuscript did not deal this in depth.

Response: Thanks for the reviewer’s suggestion. We revised this sentence. Please see it as follows.

the objectives of this review are (1) to summarize the historical process of municipal solid waste leachate treatment in China and compared the characteristics of landfills in different countries

Comment (2): Line 185: Table 6 appears before the text. Would suggest to include the table after the text.  

Response: Thanks for the reviewer’s suggestion. We adjusted the position of the table. Please see it in revised manuscript.

Comment (3): Conclusion: avoid numbering the points is the conclusion. Would suggest the length of the conclusion.

Response: Thanks for the reviewer’s suggestion. We revised the conclusion. Please see it as follows.

(1) The characteristics of the leachate are related to the production mechanism, fermentation cycle and waste composition. With the increase in landfill years, the concentration of ammonia nitrogen in the landfill leachate increases, the biodegradability decreases, and the C/N is out of balance, resulting in leachate being more difficult to treat. Compared with other countries, although the quality of landfill leachate was similar in the middle and late stages, the concentration of pollutants in the late stage leachate in China was higher, which was attributed to the differences in eating habits, landfill size and treatment methods.

(2) The leachate treatment process mainly underwent through five stages. The treatment process and the emission standard has been continuously improved. In recent years, due to the improvement of environmental protection standards and national environmental protection policies, the treatment process of landfill leachate was mainly “secondary A/O + ultrafiltration + nanofiltration (NF) + reverse osmosis (RO)” or “pretreatment + secondary A/O+MBR+Fenton Advanced Oxidation+BAF”.

(3) The future development of the landfill leachate treatment industry and the entire environmental protection industry need to meet the requirements of the Chinese social economy and urban development. Under the background of the global " Dual carbon " strategy, the green ecology of the technical route is represented by new high-efficiency denitrification, new advanced catalytic oxidation, evaporation and other fully quantitative process technologies.

Reviewer 3 Report

Dear Authors,

Overall, the study is better; there is nothing important that I feel necessary to comment on. However, I suggest you should report to Table 6 at the end of the 1st paragraph, page 5, section 3.

My best regards,

Conceição Mesquita

Author Response

We greatly appreciate the reviewer's suggestion. We revised the sentence. Please see it as follows.

From the perspective of time, leachate characteristics and treatment processes, the treatment of leachate in China was divided into the five stages, summarized in Table 6